# Hand-Engineered Image-Computable Models Can Still Outperform DNNs in V1 Similarity

**Tanish Mendki**
Department of Psychological & Brain Sciences
University of California, Santa Babara
Santa Barbara, CA 93117
tmendki@ucsb.edu

**Sudhanshu Srivastava**
Department of Cognitive Science
University of California, San Diego
San Diego, CA 92093
sus021@ucsd.edu

**Ansh Soni**
Department of Psychology
University of Pennsylvania
Philadelphia, PA 19104
anshsoni@sas.upenn.edu

## Abstract

Task-based Deep Neural Network models (DNNs) are widely used as models of inferotemporal visual cortex (IT), with early work showing a large jump over previous hand-made models [Yamins and DiCarlo, 2014]. However, recent work has suggested that over time, not only has the performance-alignment relationship plateaued but reversed, with high performing models becoming worse models of IT [Linsley et al., 2023]. Here we attempt to see if this reversal extends to earlier cortical regions. We evaluate a broad set of models, including SOTA IT similarity models [Schrimpf et al., 2020], high-task-performant CNN's and VIT's, along with other models directly attempting to model V1 [Dapello et al., 2020]. Along with DNNs we also test traditional, hand-crafted models such as HMAX [Riesenhuber and Poggio, 1999]. Surprisingly, we find that the most modern models are equal or worse models of V1 compared to HMAX even as they are better models of IT. Furthermore, HMAX becomes the best model when utilizing representational similarity scores that care about representational geometry or strict pairwise matching, being only permutation invariant. These results suggest that further research with hand-crafted image computable models is required, as these models may still outperform our most modern models in certain circumstances.

## 1 Introduction

Deep neural networks are now a standard lens for studying the visual stream. The last decade of computational neuroscience research has also established a close link between the performance of artificial neural networks and alignment with human neural responses in higher visual cortex. Early work showed that task-optimized deep neural networks (DNNs) captured important aspects of inferotemporal (IT) representations, and that increases in recognition accuracy were accompanied by gains in IT predictivity [Yamins and DiCarlo, 2014, Cadieu et al., 2014, Schrimpf et al., 2020]. This positioned the object recognition performance of networks as a useful organizing principle for emergent alignment between their features and neural activations. However, recent evaluations complicate this narrative, with Linsley et al. [2023] showing that newer networks are progressively worse models of IT responses. It is unclear if this finding extends earlier in visual cortex.

Specifically, primary visual cortex establishes the oriented and frequency tuned basis that later areas transform into tolerant, category useful representations. Unlike IT, we have hand-crafted, biologically inspired models such as HMAX that were designed to approximate these computations directly, matching a range of V1 phenomena [Riesenhuber and Poggio, 1999, Serre et al., 2007, Roos et al., 2014]. Attempting to explicitly model V1 has also continued with Dapello et al. [2020], showing that inserting an explicit V1 module in front of modern convolutional networks can improve the adversarial robustness of models without harming recognition performance [Dapello et al., 2020], highlighting the importance of early alignment.

Building on this notion, we attempt to investigate how V1 alignment has changed as we create more performant models and if they actually exceed the similarity of early hand-tuned, image-computable models. Our evaluation spans classic biologically inspired models [Riesenhuber and Poggio, 1999, Serre et al., 2007, Roos et al., 2014, Kay et al., 2008, Simoncelli and Freeman, 1995, Fukushima, 1980], early convolutional networks [Krizhevsky et al., 2012], modern convolutional networks [He et al., 2016, Liu et al., 2022], transformers [Dosovitskiy et al., 2021, Wu et al., 2021], and self-supervised or vision–language encoders [Radford et al., 2021, Mu et al., 2022], as well as architectures with an explicit V1 front end [Dapello et al., 2020] and shallow recurrent backbones [Kubilius et al., 2019]. The central result is summarized in Figure 1, where each point represents a model placed by publication year on the horizontal axis and V1 similarity on the vertical axis. The alignment envelope peaks at HMAX and does not rise with successive generations of high performing architectures. In other words, V1 alignment has not improved over time even as task accuracy and model scale have increased.

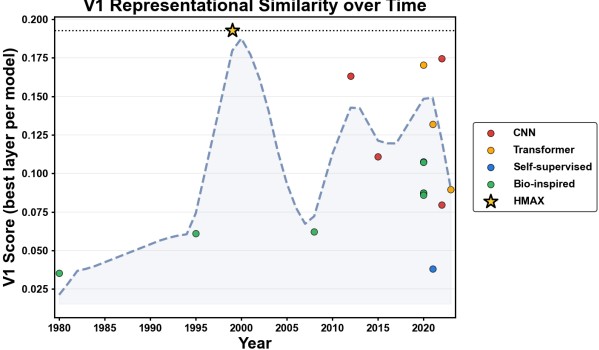

Figure 1: **V1 representational similarity has not surpassed the benchmark provided by HMAX, despite being a purely mathematical model from over two decades ago.** Each point represents a model (best V1 layer per model); colors denote families (CNN, Transformer, self-supervised, bio-inspired). The dashed curve traces a rolling average of the top models within a multi-year window. The gold star marks HMAX and the dotted line shows its score as a reference

When qualifying these results, an important angle to consider is the influence of metrics on brain similarity. The choice of metric is a known source of variability in model–brain comparisons, and different similarity functions emphasize different aspects of representation and can shift the layer to area correspondences and model rankings [Soni et al., 2024]. We therefore report multiple measures and show that the V1 stagnation pattern persists, even though absolute ranks can differ across measures. HMAX either exceeds or ties modern models for V1 representational similarity.

The following two implications are given. First, preserving early-stage fidelity appears necessary for progress in biological alignment. Models that retain V1-like computations in the front end, such as Gabor filter banks with localized pooling, remain strong references for V1 alignment [Riesenhuber and Poggio, 1999, Serre et al., 2007, Roos et al., 2014, Dapello et al., 2020]. Additionally, the results not only echo recent work in inferotemporal cortex [Linsley et al., 2023]: engineering success on recognition does not by itself yield better models of the brain. Instead, taking it a step further, we find that V1 alignment has been historically stagnant, and despite exponential gains in model performance, none of them have surpassed the representational benchmark set by HMAX.

## 2  Methods

We utilize the shared 515 subset of the Natural Scenes Dataset (NSD). This dataset contains 7T fMRI data (1.8 mm, 1.6 s) for 8 participants each viewing up to 10000 images, of which we use the 515 images all 8 participants viewed. For each subject we only extract voxels in the V1 ROI. This leads to a $515 \times N_{Voxels}$ matrix for each participant. For each model representation, we compute a correlation-based score with each of these matrices before averaging. We compute scores for all layers of each model. We define the *best layer* as the layer with the highest mean correlation across folds and participants (for the given metric). The model's representative (final) score is the mean correlation of this best layer.

We evaluated a broad model zoo spanning biologically inspired and modern vision systems: HMAX, the Gabor Wavelet Pyramid, a Steerable Pyramid representation, and a Neocognitron; standard CNNs such as AlexNet and ResNet-50; modern architectures including ViT, CvT-13, and ConvNeXt-Large; multimodal and self supervised encoders from CLIP (ConvNeXt-Large MLP) and SLIP; and multiple VOneNet variants that place a V1 front end in front of AlexNet, ResNet-50, or CORnet-S (both Gaussian and neuronal parameterizations). We also included additional ResNet-50 checkpoints from the same family. This set covers leading V1 candidates and models with explicit V1 stages.

We computed three complementary measures.

**Representational Similarity Analysis (RSA).**  For each dataset (model units and voxels), we built an $N \times N$ representational dissimilarity matrix (RDM) by measuring $1 - \text{correlation}$ between responses to every pair of images. We then measured model-brain similarity by computing Kendall's $\tau$ between the upper triangles of the two RDMs.

**Pairwise matching.**  On a training set, we match each unit in the model to one in the brain with the highest response correlation. We then evaluate on a held-out test set by computing the correlation (equivalently, the dot product of normalized response vectors. The pairwise-matching score is the average of these best-match correlations. This procedure follows [Khosla and Williams, 2024].

**Linear Predictivity.**  We fit ridge regression from model features to neural responses with five fold cross validation and a log spaced ridge grid. Features were standardized, zero variance columns were handled safely, and predictions on held out images were scored as the column-wise Pearson correlation.

## 3  Results

We evaluated V1 alignment for a broad model set that spans biologically inspired hand-crafted models, early convolutional networks, modern convolutional networks, transformers, and self-supervised encoders. For each model we selected the best V1 layer across participants and quantified alignment with three complementary measures described above. Figure 2 presents V1 alignment across the full model set using the three measures discussed earlier. Across representational similarity and pairwise matching, HMAX sits at or near the top of the distribution, with classic CNN architectures trailing closely and transformers lower still. Several VOneNet variants cluster near the middle. Linear predictivity shows a different perspective of this, with a subset of recent models including large CNNs and vision transformers as well as a CLIP trained ConvNeXt exceeding HMAX on this readout. However, these gains are modest and not universal, establishing that **a biologically inspired mathematical model such as HMAX remains a very strong contender for V1 alignment**.

We quantified differences relative to HMAX at the participant level. For each metric and model, we compared V1 scores against HMAX using paired t-tests with a Bonferroni correction at $\alpha = 0.05$. Under representational similarity, the majority of models are significantly below HMAX after correction, with a very small number being statistically indistinguishable, and none are reliably higher. Pairwise matching produces the same conclusion. Under linear predictivity, some recent CNNs and transformers show significant positive differences relative to HMAX, many models are statistically tied, and the rest are significantly below. Where advantages over HMAX appear, they are confined to the linear readout and effect sizes are small.

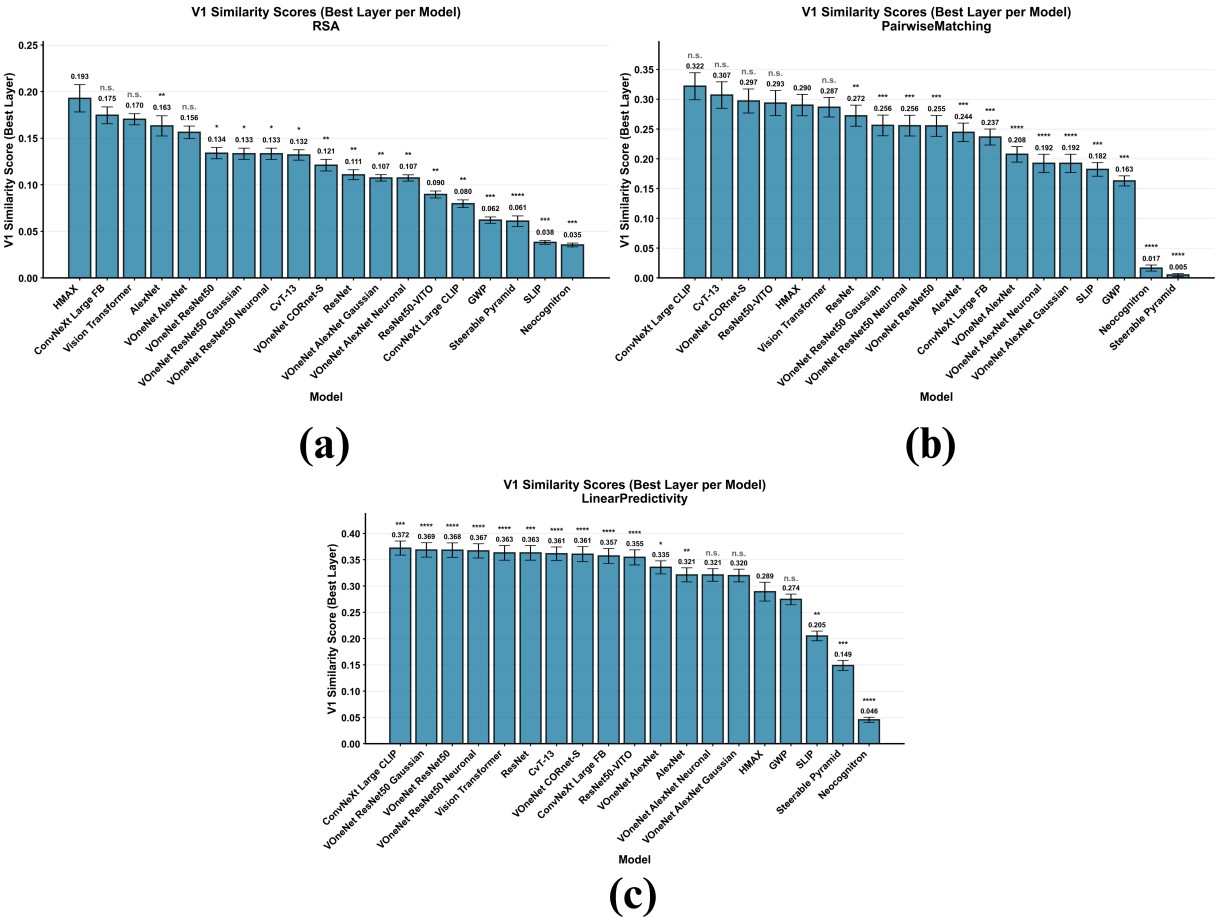

Figure 2: **Bar plots of mean V1 similarity for each model using the best V1 layer per model, with error bars showing SEM across participants and significance stars showing pairwise t-tests between models and HMAX.** Each sub-plot shows results from a different metric; (a) RSA, (b) Pairwise Matching and (c) Linear Predictivity.

Finally, we also examined how the attainable V1 alignment score has evolved over time. For representational similarity and pairwise matching, the frontier rises into the late 1990s, reaches a peak in the period anchored by HMAX, and does not move upward with the arrival of large CNNs or even transformers. Recent models densely populate the timeline but remain under the HMAX reference level. For linear predictivity, the frontier climbs through the 2010s to a value slightly above HMAX and then levels off, rather than continuing to improve.

## 4 Discussion

While early work in using DNNs for studying alignment in the visual stream suggested that task optimization led to increased alignment in IT, this claim has been contradicted by more recent work [Linsley et al., 2023]. Performance optimization then, does not seem to lead to emergent unified representations, and here we show that this gap is even more apparent in V1. The benchmark set by HMAX remains unbeaten when alignment is defined by representational geometry or by strict pairwise structure. The persistence of the HMAX benchmark suggests that key inductive biases of early vision remain underrepresented in current computational vision and image-computable models. Localized orientation and spatial frequency structure with simple pooling continue to capture V1 statistics very well, consistent with reports that adding an explicit V1 stage can improve robustness in

model performance [Dapello et al., 2020]. Preserving early-stage fidelity might still be required to lead to stronger V1 alignment.

There are limitations to consider as well. We used a best layer policy to compare models of different depths and this may hide layer specific tradeoffs. The participant pool is modest and the set of stimuli and ROIs may not cover all V1 operating regimes. Finally, publication year is an imperfect proxy for the evolution of training data, objectives, and regularization.

In sum, there is a clear historical decoupling between recognition performance and V1 alignment. Classical, mathematically explicit models set a strong baseline that remains competitive. Advancing beyond this baseline will likely require protecting early vision structure while allowing later layers to specialize, and evaluating success with multiple metrics across the hierarchy.

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
