# OpenReview forum: "Hand-Engineered Image-Computable Models Can Still Outperform DNNs in V1 Similarity"
_NeurIPS.cc/2025/Workshop/UniReps — UniReps2025_

### Official Review · Reviewer_1MF9 · 2025-09-05
**A good overview**

**Confidence:** 5

**Review:**

The paper clearly communicates the results it wants to present. However, more extensive experiments would have made the results even stronger.
Methods:
- I would like to know how the RSA was computed, especially which toolbox or implementation was used.

Future Work:
- I think future work needs to be expanded on.

Figures:
It shows that all three figures have the same trend, but the caption should mention this.

**Score:**

3

**Topic Fit:**

3

---

### Official Review · Reviewer_jPu8 · 2025-09-14
**Concise demo that learned features may not be so universal for early sensory regions**

**Confidence:** 5

**Review:**

This submission gives a concise demo that learned neural network features may not be so universal for early sensory regions. The alignment of several handcrafted features to V1 fMRI data from NSD is tested using 3 representational alignment techniques (pairwise matching, RSA, linear predictivity); their alignment outperforms optimal layers extracted from old- and new-school convolutional neural networks and transformers.

Minor concern: The result may be tied to the fact that fMRI data is used. How well does fMRI capture the response profiles of early sensory cortical neurons? Do you think the result would generalize to all the V1 benchmarks in [Brain-Score](https://www.brain-score.org/vision/leaderboard/?benchmark_regions=V1), for example, some of which use more high-fidelity recording techniques?

It would also be interesting to know _why_ exactly artificial features do worse in this context. Are their response profiles qualitatively different from hand-crafted features, unlike the universality observed between convnets and V1 ([Krizhevsky et al., 2012](https://proceedings.neurips.cc/paper_files/paper/2012/hash/c399862d3b9d6b76c8436e924a68c45b-Abstract.html), [Zeiler & Fergus, 2013](http://arxiv.org/abs/1311.2901), [Yosinski et al., 2015](http://arxiv.org/abs/1506.06579), [Sengupta et al., 2018](https://proceedings.neurips.cc/paper/2018/hash/ee14c41e92ec5c97b54cf9b74e25bd99-Abstract.html))?

**Score:**

3

**Topic Fit:**

3

---

### Official Review · Reviewer_MXC5 · 2025-09-15
**The paper challenges the assumption that scaling and task optimization inherently improve neural mechanistic models**

**Confidence:** 3

**Review:**

**Pros:**
- Use of multiple similarity metrics showing metric-dependent results like HMAX excelling in RSA and pairwise, but not always in linear predictivity.
- Benchmarked on a diverse pool of architectures of models

**Cons:**
- Relies on a relatively small stimulus set and participant pool, limiting generalizability. Using broader datasets could strengthen claims.
- The "best layer" policy and publication year proxy introduce potential biases, as noted in the limitations, which should be mitigated.

**Score:**

4

**Topic Fit:**

3

---

### Official Review · Reviewer_4Lfc · 2025-09-16
**Interesting preliminary work**

**Confidence:** 5

**Review:**

This paper explores a very interesting topic: whether the "performance -> brain-alignment" story extends to early visual cortex (NSD). The authors compare a broad model zoo and use three brain-similarity measures, finding that the hand-engineered HMAX outperforms other models on representational similarity. The authors conclude that conclude that early-vision inductive biases remain underrepresented in current DNNs.

The paper is certainly relevant for UniReps and contributes to a relevant ongoing discussion about metrics. The model coverage is good for a workshop.

Some general comments: while using the NSD 515 split is fine here for UniReps, it is important (and not too difficult) to evaluate this on other brain datasets. Conclusions about V1 geometry are constrained by fMRI resolution, and the evaluation should extend in future work to other publicly available datasets (especially from nonhuman primates/single neuron recordings, but also NSD synthetic, etc). Also, I believe this was stated in the paper (but just want to clarify), when stating you choose the layer by the best score, what is this score? My understanding is that a different layer is chosen for different evaluations, and that layer choice is based on the evaluation metric (i.e., you use the best cross-validated layer in terms of predictivity, when doing the predictivity analyses). Also, I'd be very curious to see future work exploring more theoretical justifications regarding why HMAX aligns the most by representational geometry.

**Score:**

4

**Topic Fit:**

3